# Effect of the Biodegradable Component Addition to the Molding Sand on the Microstructure and Properties of Ductile Iron Castings

**DOI:** 10.3390/ma15041552

**Published:** 2022-02-18

**Authors:** Katarzyna Major-Gabryś, Małgorzata Hosadyna-Kondracka, Adelajda Polkowska, Małgorzata Warmuzek

**Affiliations:** 1Faculty of Foundry Engineering, AGH University of Science and Technology, Mickiewicza 30, 30-059 Krakow, Poland; katmg@agh.edu.pl; 2Łukasiewicz Research Network-Krakow Institute of Technology, Zakopiańska 73, 30-418 Krakow, Poland; adelajda.polkowska@kit.lukasiewicz.gov.pl (A.P.); malgorzata.warmuzek@kit.lukasiewicz.gov.pl (M.W.)

**Keywords:** metal casting, ductile cast iron, molding sand, organic binder, biodegradable additive

## Abstract

In this work, the results of the examinations of the effect of the mold material and mold technology on the microstructure and properties of the casts parts of ductile cast iron have been presented. Four different self-hardening molding sands based on fresh silica sand from Grudzen Las, with organic binders (no-bake process), were used to prepare molds for tested castings. A novelty is the use of molding sand with a two-component binder: furfuryl resin-polycaprolactone PCL biomaterial. The molds were poured with ductile iron according to standard PN-EN 1563:2018-10. The microstructure of the experimental castings was examined on metallographic cross-sections with PN-EN ISO 945-1:2019-09 standard. Observations were made in the area at the casting/mold boundary and in a zone approximately 10 mm from the surface of the casting with a light microscope. The tensile test at room temperature was conducted according to standard PN-EN ISO 6892-1:2016-09. Circular cross-section test pieces, machined from samples taken from castings, were used. In the present experiment, it was stated that interactions between the mold material of different compositions and liquid cast iron at the stage of casting solidification led to some evolution of casting’s microstructure in the superficial layer, such as a pearlite rim observed for acidic mold sand, a ferritic rim for alkaline sand, and graphite spheroids degeneration, especially spectacular for the acidic mold with polycaprolactone (PCL) addition. These microstructural effects may point to the interference of the direct chemical interactions between liquid alloy and the components released from the mold sand, such as sulfur and oxygen. Particularly noteworthy is the observation that the use of molding sand with furfuryl resin with the addition of biodegradable PCL material does not lead to an unfavorable modification of the mechanical properties in the casting. The samples taken from Casting No. 2, made on the acidic molding sand with the participation of biodegradable material, had an average strength of 672 MPa, the highest average strength UTS-among all tested molding sands. However, the elongation after fracture was 48% lower compared to the reference samples from Casting No. 1 from the sand without the addition of PCL.

## 1. Introduction

Cast iron accounts for about 70% of the total production of castings, which confirms that it is the most used alloy in the industry [1]. Cast iron castings are widely used in the automotive industry, in sea and rail transport, in the energy industry, agriculture, and construction, as products that do not carry loads, mainly of gray cast iron (cylinders, ingot molds, pistons); more loaded castings of malleable and nodular cast iron (parts of agricultural machinery, car parts, fittings, camshafts, crankshafts, gears, machine tool spindles), as well as castings with the high abrasion resistance of white cast iron (mill balls, brake pads). The advantages of cast iron castings are high durability and strength, resistance to abrasion, good ability to dampen vibrations, the possibility of obtaining various and complicated shapes of castings as well as low production costs. The production of a large-size casting with a complex shape, characterized by high quality while maintaining the required functional properties, involves many stages of the production process. These stages are interdependent and ultimately determine the quality and competitiveness of the manufactured elements. The main stages of the large-size castings manufacturing process include:−The selection of the chemical composition of the casting;−The proper selection of mold and core technology;−The preparation, smelting and refining of the liquid alloy (modifiers);−The post-casting treatment, i.e., removal of gating systems, treatment heat, finishing (polishing, shot blasting) of the raw surface of castings;−The quality control of the finished product.

This paper presents the impact of the type of molding sand used in the mold technology on the properties of the produced iron casting. Four different molding and core sands, dedicated to the production of large-size castings, were selected for the tests. They are molding sands with organic binders. Furthermore, in one molding sand, the addition of biodegradable material to the binder was used in order to accelerate the process of biodegradation of the binder waste remaining after the casting process. The authors’ earlier studies proved an increased biodegradability of post-production residues of the binding material in the case of using a new two-component binder in relation to the analogous molding sand without the additive [2]. This approach to the issue of improving the ecology of the process is particularly important in the case of large-size castings production, where the amount of waste binding material is large.

The rapid increase recently observed in the production of heavy castings of cast iron, both vermicular and nodular, also places special demands on the quality of molding materials. While molding sands with furfuryl resins are very well suited for the production of small and medium-sized castings, their use for the production of heavy castings will require significant technological modifications. Thus, the constant improvement of molding technologies is still an important challenge.

The latest trends in foundry technologies focus on improving the dimensional accuracy and functional quality of castings and on meeting the environmental protection requirements introduced in the European Union. Subsequent generations of molding materials are elaborated, including their second generation based on sands bound with various binders, which can also be used for the production of both foundry molds and cores [3].

The second-generation molding technologies mainly use organic binders based on synthetic resins, but the inorganic ones based on hydrated sodium silicate can also be used [3,4,5]. At present, molding technologies with the use of alkyd resins hardened by catalyst based on isocyanates and alkaline phenolic resin hardened by esters are becoming a serious alternative in the heavy steel castings industry, mainly due to their less harmful environmental impact [5]. Among the basic advantages of the molding sands with alkyd binder, there is [3]: the possibility of using a very high proportional fraction of reclaim in the mixture (up to 90%); high plasticity of the sand during contraction of the steel casting (as opposed to phenolic and furfuryl resins); no nitrogen content, no addition of formaldehyde and water to their composition; a lower amount of gases emitted during pouring of molten metal and low level of the toxic compounds formed during mixing and setting of sands (as opposed to furfuryl resins). Unfortunately, using more coarse sands requires the application of protective coatings (hydrous or alcoholic).

Another group of materials for second-generation molds are those based on loose self-hardening sands with furfuryl resins. These are materials based on grains of silica sand and/or their reclaim, with furfuryl resin and an acid hardener, such as organic sulfonic acids, sometimes in combination with orthophosphoric acid, and inorganic sulfuric or phosphoric acids. The silica sand used for these materials should contain a maximum of 0.2% of the clay and its pH should be close to 7 [6].

Since the time of solidification and cooling of heavy castings, from the moment the liquid metal is poured into the mold, takes up to several days, local interactions between the mold material and the liquid alloy, and then the cooling-down casting may affect its properties to a much more significant extent than in the case of small castings [7,8]. Especially in the case of using molds made of self-hardening sands with furfuryl resins hardened with acid hardeners, their negative influence on the graphite spheroidization process in the casting surface layer was observed. This phenomenon depends on the sulfur content in the molding or core sands and the conditions of the oxidation process of the solidifying surface casting [8,9,10,11,12]. The sulfur-containing gases released from the molds materials are absorbed by the liquid metal. Therefore, a reduction in the effectiveness of the magnesium modifier is observed due to the formation of magnesium sulfide. Sulfur saturation of surface zones of castings made in molds of loose self-hardening molding/core sands with furfuryl resins is also observed in steel castings [10,11]. The effects of the interaction between the cast iron and the mold material are manifested as casting skin, including roughness of the casting surface and other microstructural effects, is considered to be unfavorable [6,8,9].

According to the tendencies observed in recent years, molding processes must meet high requirements connected to environmental protection including problems related to the disposal of waste from used molding materials [1,13].

The idea of K. Major-Gabryś [1] is to introduce an additive of biodegradable material to the commercial organic binder. The trend causing a gradual replacement of binding materials produced from a petrochemical origin with biomaterials is observed in foundry technologies development [14,15,16,17]. In the 90′s General Motors Co. (Detroit, MI, USA) elaborated a new binding system based on protein composition GMBOND^TM^ [14]. The materials in the protein binder come from natural renewable resources and consist of polypeptide chains, which are ecologically friendly. The binder is well soluble in water and the binding process begins during the dehydration reaction of wet molding sand. The technological experiments using the protein binder showed good enough castings properties, though the binder quantity was reduced by 45% in comparison to the furfuryl resin binder quantity used in core production by hot-box technology [14]. Another example of using biodegradable materials in molding sands technologies was proposed by K. Rusin’s scientific group [15]. The usage of biogenic binders based on proteins obtained from by-products of pharmacy industry production was tested. The binders were water soluble non-toxic polymers, including different polypeptide molecules with long amino acids chains. There is no chemical reaction in the process, only the reaction of dehydration during the heating process. This kind of molding sand can be used for light alloys like aluminum castings [15]. B. Grabowska [16,17] proposed the usage of aqueous biodegradable polymeric compositions consisting of acrylic derivatives and modified natural polymers as foundry molding sands’ binders (BioCo binders). Molding sands with BioCo binders can be used in iron castings production [17]. K. Major-Gabryś investigated the possibility of using biodegradable materials, such as PLA, PHB, or PCL as binders for molding sands production. Research results showed lower toxicity and greater capacity for mechanical reclamation of molding sands with biodegradable materials as binders [1].

The idea of improving molding sand presented in this paper is to introduce an additive of biomaterial to the commercial organic binder. Therefore, molding sand with a two-component binder: furfuryl resin and biodegradable PCL, will be also used and the influence of the biodegradable additive on changes in the microstructure and properties of the casting will be tested.

Literature data [18,19,20,21,22] shows the possibility to use biomaterials as additives to petroleum binders in order to cause biodegradation of materials from the petrochemical industry. Various synthetic resins can be fragmentized and biologically assimilated, however, most of these processes could take tens or even hundreds of years. According to research results of G. Scott [19], one of the solutions for this problem is a partial replacement of the resins with oxy-biodegradable polymers characterized with short decomposition time. Oxy-biodegradation of polymers is possible, thanks to special pro-oxidant additives which are usually compounds of iron, nickel, cobalt, or manganese together with carefully formulated stabilizers [19]. These additives can dissociate the bonding between carbon atoms. An example is PCL (polycaprolactone), which is compatible with many other polymers. It is partially compatible or mechanically compatible with polymers, such as polyvinyl acetate (PVAc), polystyrene (PS), polycarbonate, etc., and with other polymers, such as polyvinyl chloride (PVC), styrene acrylonitrile copolymer (SAN), poly (hydroxy ether), etc. This feature of polycaprolactone enables the formation of various “bio-destructive” mixtures using it as a biodegradable component [20]. Initial studies of “bio-destructive” polymer blends using PCL as a biodegradable component relate to a PC polyolefin blend system based on polyolefins, such as LDPE (low density polyethylene) and PP (polypropylene). More extensive studies on the biodegradability of PCL/polyolefin blends, including the relationship between biodegradability and phase structure, have been presented by A. Iwamoto and Y. Tokiwa [23].

The main objective of the research presented in the article was to analyze the microstructural effects related to the interaction with the mold material in ductile iron castings, in the surface layer, and inside the cross-section, depending on the type of molding material, with acid and alkaline hardeners. The characteristics of the matrix microstructure and graphite morphology in two zones on the casting cross-section, near the surface and inside the casting, were determined to distinguish and estimate the specific effects of the interaction of the liquid metal with the tested molding materials.

## 2. Materials and Methods

The molding mixtures based on fresh silica sand from Grudzen Las (Slawno, Poland) were characterized by the following parameters: granulation 0.20/0.32/0.40; d50 = 0.31 mm; pH = 7. The composition of the molding mixtures is presented in Table 1. The molding materials were prepared in a laboratory mixer LM-R1, using mixing times: sand and hardener—60 s and sand, hardener and binder—50 s.

To reveal the possible effects of metal/mold interactions, the mold design, as well as sampling area, were determined based on literature data from Linke and Sluis [12]. The dimensions of the experimental casting were 195 mm × 175 mm × 120 mm. The characteristic U-shape allowed obtaining the highest possible concentration of gas released from the mold material. A scheme of the casting is shown in Figure 1. Standard tensile tests with circular cross-section test pieces and microstructure observations were carried out for test samples taken from the experimental castings in the area of the assumed intense interactions between mold and casting surface.

Experimental castings were made of ductile cast iron according to standard PN-EN 1563:2018-10 “Founding-Spheroidal graphite cast irons” [25]. The chemical composition was 3.30% C, 2.61% Si, 0.38% Mn, 0.04% P, 0.01% S, 0.06% Cr, 0.07% Ni, 0.06% Mg, 0.07% Cu, 0.02% V, 0.02% Al, 0.01% Ti and the rest-Fe (as estimated by optical emission spectrometry method, ARL MA spectrometer (Thermo Scientific). Metal melting was carried out in the Radyne furnace AMF 45/150 of medium frequency induction furnace with a crucible of 100 kg capacity of charge and neutral liner. Spheroidization and modification were performed in a slender ladle, using a FeSiMg9 modifier (Elkem, Norway)

The molds were poured with ductile iron with a drain temperature of about 1420 °C and a pouring temperature of about 1370 °C. Figure 2 includes photographs of the examined molds, produced from molding sands with composition presented in Table 1.

The microstructure of the experimental castings was examined on metallographic cross-sections, etched with 4% Nital reagent (POCH, Gliwice, Poland), under Zeiss Axio Observer Z1m light microscope (Jena, Germany). Observations were made in the area at the casting/mold boundary and in a zone approximately 10 mm from the surface of the casting. Quantitative microstructure characteristics were carried out utilizing the commercial system for analysis of the microscope images Axio Vision (version 4.8.2.0, Zeiss, Jena, Germany).

The tensile test at room temperature was conducted according to standard PN-EN ISO 6892-1:2016-09, method B [24]. Circular cross-section test pieces of 14 mm diameter (drawing and dimensions on Figure 1), machined from samples taken from castings, were used. Three pieces were tested for each casting. Examinations were carried out on EU-20 strength testing machine (VEB, Leipzig, Germany) with a range of 0–200 kN. The strain rate for tensile testing was 16 MPa/s. The percentage elongation after fracture A was calculated from Equation (1) [24]:(1)$$A=\frac{L_{u}\;–L_{o}\;}{L_{o}\;}\cdot 100\%$$

*L_o_* the original gauge length;

*L_u_* is the final gauge length after fracture.

## 3. Results and Discussion

The photos of the experimental castings (Figure 3a) and the area with traces of interaction of the liquid metal with the mold surface, remain on the casting surface (Figure 3b). An area of increased roughness appeared on the surface of all the examined castings, reported previously as a typical effect for such molding technology. Hence, it is necessary to estimate the range of interactions in the material of the tested casting.

### 3.1. Experimental Castings Microstructure

The microstructure of the ductile cast iron formed in the examined castings is presented in Figure 4, Figure 5, Figure 6 and Figure 7. The observed morphology and phase composition of the microstructure is typical for ductile iron according to [25] with the chemical composition used for testing castings, i.e., pearlitic–ferritic matrix with graphite spheroids. Ferrite forms characteristic envelopes around graphite spheroids. Graphite nodules observed in these castings have been ascribed to classes V and VI (Figure 8), according to designation in standard PN-EN ISO 945-1:2019-09 [26]. Nevertheless, a variation in microstructural effects was observed, related to the type of material used for the casting molds.

The revealed microstructural effects were observed in two zones of wall section: in the surface layer (casting skin) and the area of 10 mm from the surface. The description of the microstructural evolution in these zones concerned the local phase composition of the metal matrix and the shape of graphite.

In castings No. 1 and No. 2, which were both made of acid molding sand molds, in the matrix microstructure in the surface layer (skin), different effects were found. In Casting No. 1 typical pearlite skin (about 5 µm) was revealed as reported previously in [10]. The layer of material with degenerate graphite nodules (classes IV and V) was thicker (approx. 200 µm, Figure 4). In Casting No. 2, from mold No. 2, in which part of the hardener was replaced with PCL polycaprolactone, no visible pearlite skin was observed. In the superficial layer, thicker than in Casting No. 1 (approx. 500 µm), there was degenerated graphite in flake form (class I, Figure 5). In castings No. 3 and No. 4, made in molds of alkaline materials, the main microstructural effect in the surface layer was decarburization, i.e., reduction in the volume fraction of pearlite (Figure 6 and Figure 7; Table 2). However, while in Casting No. 3, degenerate graphite was visible in the superficial layer (class IV and V, Figure 6), in the superficial layer the Casting No. 4, along with the almost complete ferritization of the matrix (Figure 7, Table 2), also the graphite particles disappeared (Figure 7). The thickness of the casting skin in castings No. 3 and No. 4 was about 500 µm, which can be compared with the result given in [6,10] for castings produced in molds with a protective coating.

The observed changes in the matrix microstructure can be considered by the effect of the superposition of the diffusion of the mold components into the liquid alloy and its oxidation. The formation of the thin pearlitic rim in Casting No. 1 can be explained by the enrichment of the alloy with sulfur and carbon coming from the mold material. On the other hand, the oxygen produced by the contact of the hot metal with the mold leads to the oxidation of the carbon from liquid alloy. This may result in such a degree of carbon depletion of the alloy on the mold/casting interface that a completely decarburized layer is formed, observed in Casting No. 4 (Figure 7) as a ferritic rim, without graphite particles. The degeneration of the shape of graphite nodules in the casting skin can be explained by the bonding of Mg from the used modifier with sulfur released from the mold material as described in [5,7,10]. Although graphite degeneration was found in castings No. 1, 2 and 3, it was most intense in Casting No. 3 where flake graphite appeared. Magnesium oxidation can be considered as an additional factor contributing to the weakening of the spheroidization effect, observed in the examined castings [7,10].

Microstructural effects in the material inside the examined castings were recognized as changes in the metal matrix phase composition (Figure 4, Figure 5, Figure 6 and Figure 7) and morphology of the graphite nodules (Figure 8 and Figure 9). The ratio V_vF_/V_vP_ of volume fraction of ferrite V_vF_ to perlite V_vP_ in the examined castings was different (Table 2). These microstructural differences can be attributed to the actual temperature field created during the solidification of the casting, which is subordinate to the thermal conductivity of the mold material. However, one should also take into account the possibility of decarbonization of the liquid alloy in the depths of the casting, as a result of long-term in situ interactions with the mold surface and with the released gaseous reaction products. Although the visual analysis did not reveal differences in the morphology of the graphite nodules, especially between castings 1, 2 and 3, more detailed image analysis allowed for their identification. The nodule size distribution (Figure 9) and roundness coefficient R (R = 4 Aa/(πFD_max_)^2^, R = 0–1), where: Aa—the area of particle cross-section, FD_max_—the value of maximum Feret diameter, for the randomly chosen population of graphite particles (Table 2), showed the effect of molding sand composition on the morphological characteristics of the graphite spheroids. 

In the castings No. 1 and No. 2, made in acidic molds, graphite nodules were larger than those in the castings made in alkyd molds (castings No. 3 and 4, Figure 9, Table 2). Although in the superficial layers an important difference in graphite morphology was observed, in material inside all the examined castings, the roundness of graphite nodules was similar. In castings No. 1, 2, 3, the volume fraction of nodules with roundness R > 0.5 was 0.6 and only in specimen No. 4 slightly lower than 0.5. Nevertheless, graphite nodules populations in the examined specimens differ not only by their mean size, defined here as max ECD (equilibrium circle diameter, Figure 9) and FD max (Table 2) but also by distribution of the size of the nodule, visible in microstructure images (Figure 8). In castings No. 1, 2, 3 (Figure 9a–c) distribution of the graphite nodule size is typical for that observed usually in ductile cast iron, i.e., there is one fraction of large spheroids of similar diameter and the second one, more numerous–of very dispersed particles, of ECD < 10 µm. Casting No. 4 Is dominated by the fraction of small graphite particles with an ECD < 10 µm (Figure 8d and Figure 9d). Thus, based on the results of microscopic observations, it was shown that the microstructure of the metal matrix and the morphology of graphite in Casting No. 4, made in a mold with an alkaline phenolic resin, differed significantly from those revealed in other castings, both in the surface layer and inside the casting. Morphological characteristics of the microstructure observed in the experimental castings were summarized in Table 2.

### 3.2. Results of the Standard Tensile Test

The mechanical properties measured for the examined specimens taken from the experimental castings are presented in Table 3. The results are also graphically depicted in Figure 10. The obtained results, in reference to the standard [25], indicate the material designation of EN-GJS-500-7C with relevant wall thickness in the range of 60–200 mm. The minimal 0.2% proof strength should be 260 MPa, tensile strength 400 MPa and elongation after fracture 3%. The samples taken from Casting No. 2, made on the acidic molding sand with the participation of biodegradable material, had the highest average strength UTS—among all tested molding sands—of 672 MPa. However, the elongation after fracture was 48% lower compared to the reference samples from Casting No. 1 from the sand without the addition of PCL. It can be observed that samples from acidic molding sands have higher UTS strength at the level 656–672 MPa than those from alkali molding sands with 542–589 MPa.

The results of the standard tensile test signalize that the measured mechanical properties were influenced by the characteristics of the microstructure, and thus, indirectly also by used mold materials. Two types of material effects can be indicated: increased material strength in castings No. 1 and 2 made in molds with an acid hardener compared to that obtained in castings No. 3 and 4, made in molds based on alkali resins, increased ductility of the material in castings No. 3 and 4. (Table 1, Table 2 and Table 3). Simultaneously, these effects can be related to microstructure characteristics. The measured increase in UTS can be assigned to the common effect of the lower ferrite fraction (V_vF_) and larger graphite nodules, while increased elongation to fracture–to the effect of higher ferrite fraction (V_vF_) and smaller graphite nodules. In each group of castings (1. made in molds with acid hardener and 2. made in mold based on alkaline resin), some increase in yield strength value was related to the presence of smaller graphite nodules (Table 2 and Table 3). Hence, the influence of graphite morphology should be assumed, although the studies conducted so far indicate a weak relationship between the size of spheroids and the yield stress. The microstructural determination of the mechanical properties of ductile iron has not been identified so far as reported in Refs. [27,28,29]; therefore, the results obtained in this study should be considered as the initial stage of research.

## 4. Conclusions

Based on the analysis of the results of the examination carried out in this work the following conclusions can be formulated: The microstructural effects in the tested castings were revealed in two zones, depending on the area of observation: in the surface layer (casting skin) and in the zone approximately 10 mm from the surface of the casting. These effects manifested themselves through changes in the phase composition of the metal matrix and the shape of graphite spheroids. The effects of the superficial zone on mechanical properties can be considered in two aspects, related to a. Material and b. Cast part.a.Material effects result from the diffusive exchange of elements at the metal/form interface. The resulting concentration gradient of individual components in the liquid alloy affects the final image of the microstructure of the material. This impact concerns both the matrix composition and graphite morphology, and then the observed differentiation of properties (Table 2 and Table 3).b.Different ranges of superficial zone of different matrix and graphite morphology observed in the examined cast parts can influence on useful properties of cast parts during specific exploitation conditions.Experimental castings were characterized by a different thickness and microstructure of the surface layer. The microstructural effects in the superficial layer, such as the pearlite rim observed for acidic mold sand, the ferritic rim for alkaline, and graphite spheroids degeneration, especially spectacular for acidic mold with PCL addition, may point to the interference of the chemical interactions between liquid alloy and components released from mold sand, such as sulfur and oxygen.The microstructural effects observed on the cross-section of the examined castings, in 10 mm from the surface, such as a change in the volume fraction of ferrite in metal matrix and changes in the morphological characteristics of graphite spheroids, indicate the possibility of a long-term, indirect influence of the used mold materials, modified by adding various binders and hardener fractions to silica sand (Table 1).Two types of material effects can be indicated: 1. increased material strength in castings No. 1 and 2 made in molds with an acid hardener compared to that obtained in castings No. 3 and 4, made in molds based on alkali resins, and 2. increased ductility of the material in castings No. 3 and 4. The obtained results indicate the possibility of optimizing the composition of the mold material, i.e., the selection of additives used as binders and hardeners to control the mechanical properties of the casting.Quantitative image analysis revealed some differences in graphite morphology, not always accounted for in the visual analysis results. From a comparison of the histograms in Figure 9b, it follows that the spheroidal size distribution is bimodal, i.e., there are probably two sets of nodules: in sample 2 (Figure 9b) the first with a diameter of 120–140 mm and the second with a diameter of 30–50 mm, while in sample 3 (Figure 9c)—the first with a diameter of 90–100 mm and the second, probably not very numerous, with a diameter of about 130 mm. This difference in the size distribution of spheroids can be considered a factor of the synergy effect of both components of the microstructure, i.e., graphite and matrix, on the mechanical properties (Table 3).At this stage of the research, it can be assumed that the difference in the matrix microstructure results indirectly from:a.Direct diffusion interactions at the metal/mold interface, which are influenced by the composition of the molding sand,b.The temperature field occurring in the casting at the subsequent stages of solidification, when the microstructure is formed, is determined by the thermophysical properties of the mold depending on the components used. As these presented results concern of initial stage of research, these phenomena are still under examination.

## Figures and Tables

**Figure 1 materials-15-01552-f001:**
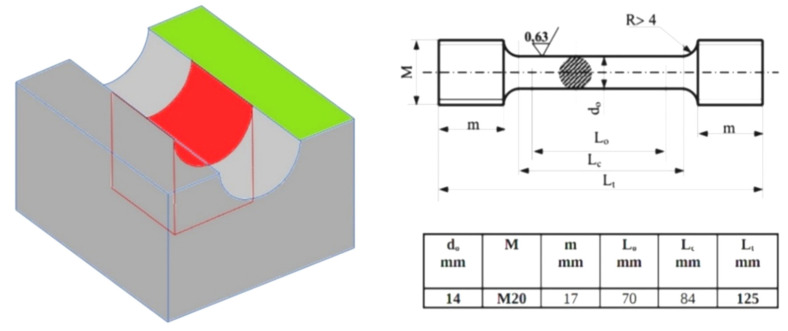
The scheme of the experimental casting with marked sampling site for metallographic (red area) and mechanical properties (green mark) examinations; drawing and dimensions of circular cross-section test pieces on right [5,12,24].

**Figure 2 materials-15-01552-f002:**
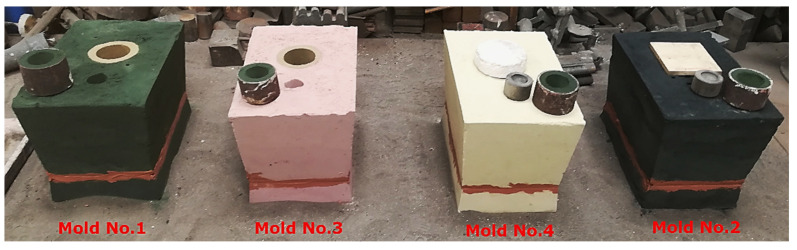
Molds used for experimental castings.

**Figure 3 materials-15-01552-f003:**
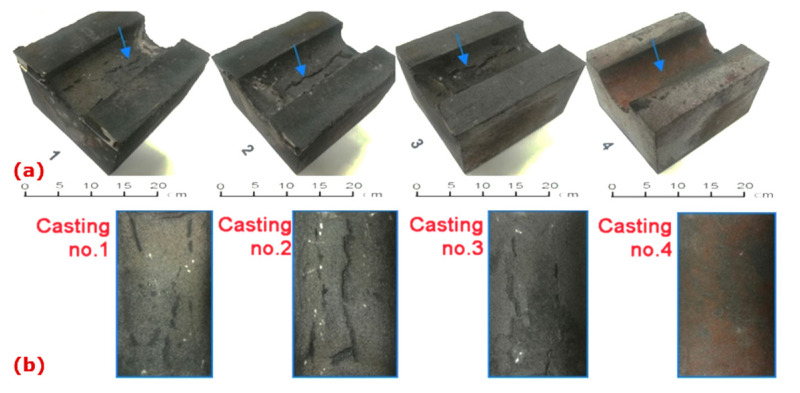
(**a**) Experimental castings 195 mm × 175 mm × 120 mm; area at interface mold/casting (blue arrows) (**b**) indicated areas with traces of interaction liquid metal/mold.

**Figure 4 materials-15-01552-f004:**
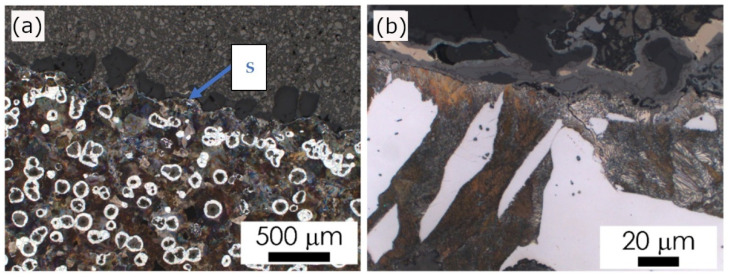
Casting No. 1, prepared in acidic molding sand No. 1, LM etched with 4% Nital; (**a**) microstructure of alloy, microscope magnification 100×, (**b**) microstructure of the superficial zone (S), microscope magnification 1000×.

**Figure 5 materials-15-01552-f005:**
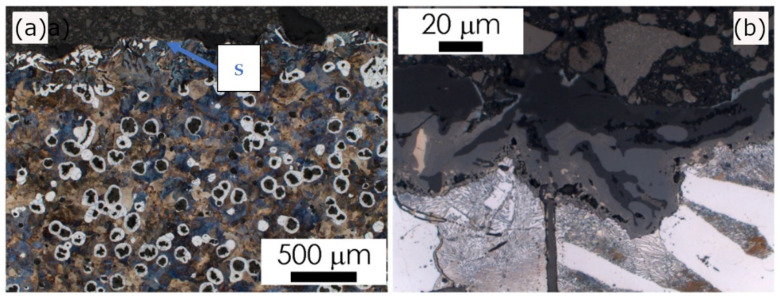
Casting No. 2, prepared in acidic molding sand No. 2, LM etched with 4% Nital; (**a**) microstructure of alloy, microscope magnification 100×, (**b**) microstructure of the superficial zone (S), microscope magnification 1000×.

**Figure 6 materials-15-01552-f006:**
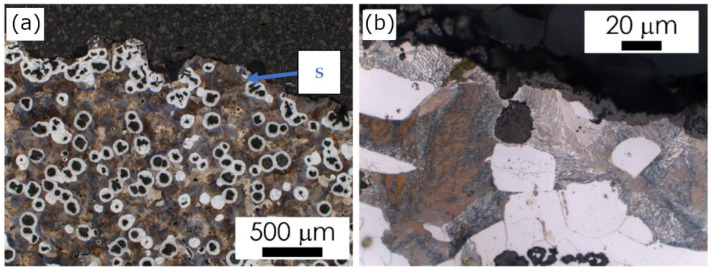
Casting No. 3, prepared in alkyd molding sand No. 3, LM etched with 4% Nital; (**a**) microstructure of alloy, microscope magnification 100,100×, (**b**) microstructure of the superficial zone (S), microscope magnification 1000×.

**Figure 7 materials-15-01552-f007:**
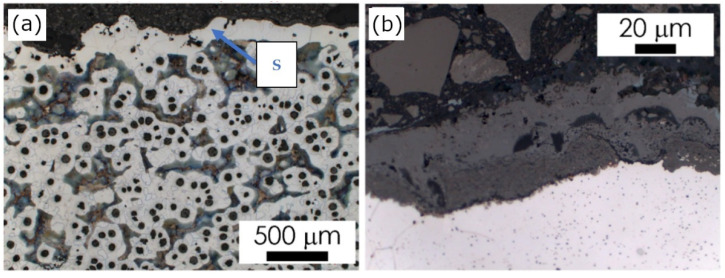
Casting No. 4, prepared in alkaline molding sand No. 4, LM etched with 4% Nital; (**a**) microstructure of alloy, microscope magnification 100×, (**b**) microstructure of the superficial zone (S), microscope magnification 1000×.

**Figure 8 materials-15-01552-f008:**
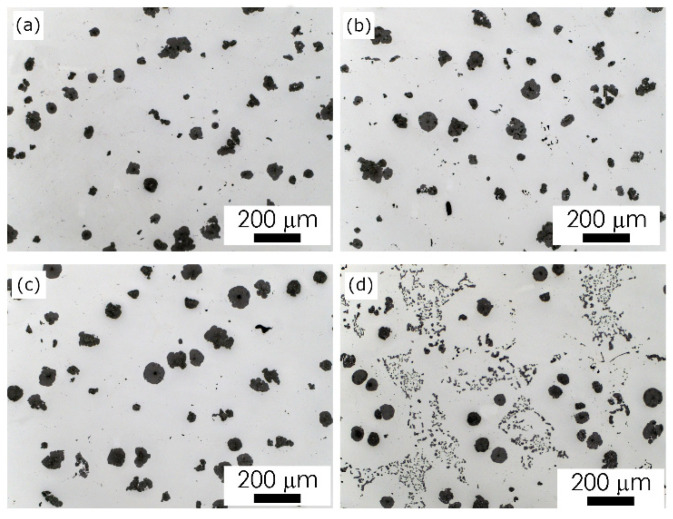
Graphite nodules morphology in the examined castings, LM, non-etched cross sections, microscope magnification 100×; (**a**) Casting No. 1 (acidic molding sand No. 1), (**b**) Casting No. 2 (acidic molding sand No. 2), (**c**) Casting No. 3 (alkyd molding sand No. 3), (**d**) Casting No. 4 (alkaline molding sand No. 4).

**Figure 9 materials-15-01552-f009:**
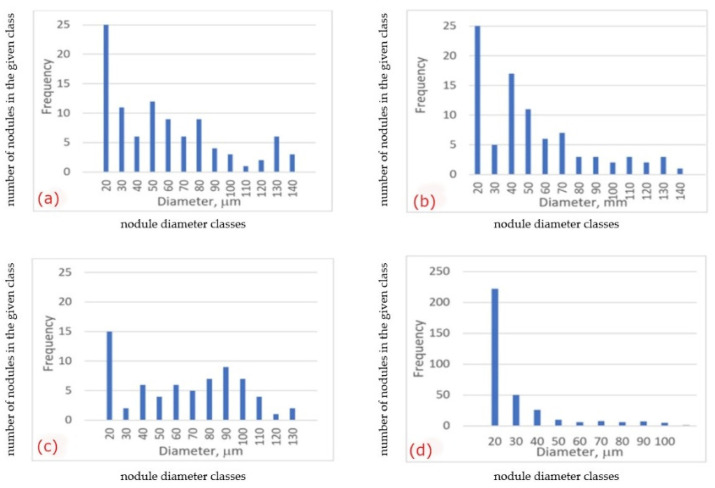
Distribution of the ECD of the graphite nodules in the examined castings; (**a**) Casting No. 1 (acidic molding sand No. 1), (**b**) Casting No. 2 (acidic molding sand No. 2), (**c**) Casting No. 3 (alkyd molding sand No. 3), (**d**) Casting No. 4 (alkaline molding sand No. 4).

**Figure 10 materials-15-01552-f010:**
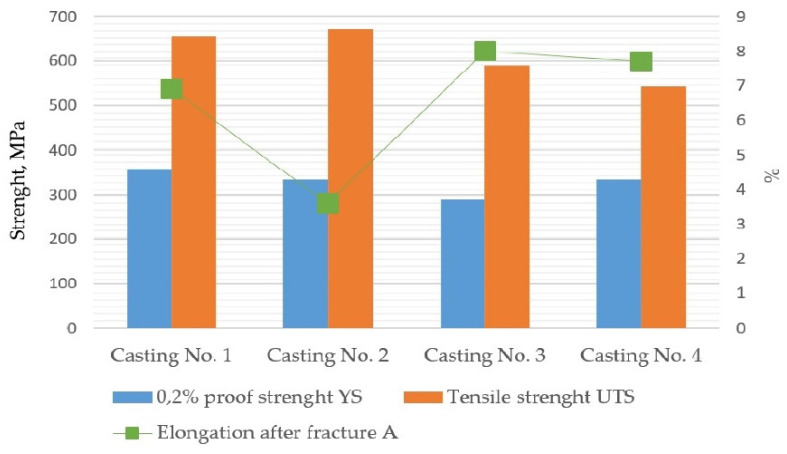
The mechanical properties of the tested samples obtained in the tensile test.

**Table 1 materials-15-01552-t001:** The composition of molding sands used in the test.

Molding sand No. 1	silica sand	100 parts by mass	
	furfuryl-resole resin	1.1 parts by mass	
	hardener-a mixture of sulfuric acids in an aqueous solution	0.55 parts by mass	
Molding sand No. 2	silica sand	100 parts by mass	
	furfuryl-resole resin	1.1 parts by mass	95%
	PCL polycaprolactone		5%
	hardener-a mixture of sulfuric acids in an aqueous solution	0.523 parts by mass	
Molding sand No. 3	silica sand	100 parts by mass	
	alkyd resin	1.1 parts by mass	
	hardener-catalyst based on isocyanates	0.275 parts by mass	
Molding sand No. 4	silica sand	100 parts by mass	
	alkaline phenolic resin	1.1 parts by mass	
	hardener-esters	0.22 parts by mass	

**Table 2 materials-15-01552-t002:** Morphological characteristics of the microstructure in the experimental castings.

	Specimen No. 1	Specimen No. 2	Specimen No. 3	Specimen No. 4
Microstructure effects in the superficial layer	Despheroidization, Pearlite skin	Despheroidization	Decarburization, Ferrite skin-local despheroidization	Decarburization, Ferrite skin depleted in graphite
Fraction of spheroids of roundness > 0.5	0.2	0.1	0.3	-
V_vF_	0.1	0.3	0.7	1.0
Microstructure in casting	Pearlite and ferrite	Pearlite and ferrite	Pearlite and ferrite	Pearlite and ferrite
V_vF_	0.2	0.1	0.2	0.8
ECD, µm	132	140	126	103
max FD, µm	187	192	166	132
Fraction of spheroids of roundness > 0.5	0.6	0.6	0.6	0.5

V_vF_ volume fraction of ferrite; ECD equilibrium circle diameter; max FD value of maximum Feret (mean value).

**Table 3 materials-15-01552-t003:** Tensile test results of the specimens from the examined castings (average of 3 results).

	0.2% Proof Strenght YS, MPa	Tensile Strenght UTS, MPa	Elongation after Fracture A, %
Casting No. 1	356 (SD = 17)	656 (SD = 15)	6.9 (SD = 1.6)
Casting No. 2	333 (SD = 6)	672 (SD = 18)	3.6 (SD = 1.2)
Casting No. 3	290 (SD = 4)	589 (SD = 24)	8.0 (SD = 2.1)
Casting No. 4	334 (SD = 5)	542 (SD = 9)	7.7 (SD = 2.7)

SD—standard deviation.

## Data Availability

Data are contained within the article.

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
