# Peer review of "Effect of the Biodegradable Component Addition to the Molding Sand on the Microstructure and Properties of Ductile Iron Castings"

_materials, 2022, doi:10.3390/ma15041552_

Round 1
Reviewer 1 Report
Thank you for submitting your paper. The work done here draws attention to a significant subject in ductile cast iron. I have found the paper to be interesting. However, several issues need to be addressed properly before the paper is being considered for publication. My comments including major and minor concerns are given below:
- Please consider reviewing the abstract and highlight the novelty, major findings, and conclusions. I suggest reorganizing the abstract, highlighting the novelties introduced. The abstract should contain answers to the following questions:
- What problem was studied and why is it important?
- What methods were used?
- What conclusions can be drawn from the results? (Please provide specific results and not generic ones).
- The abstract must be improved. Please use numbers or % terms to clearly shows us the results in your experimental work. Please expand the abstract.
- Please consider reporting on studies related to your work from mdpi journals.
- The introduction must be expanded, please consider improving the introduction, provide more in-depth critical review about past studies similar to your work, mention what they did and what were their main findings then highlight how does your current study brings new difference to the field.
- The abstract should be on paragraph.
- Please combine small paragraphs into larger ones, I can see many of them in the manuscript especially in the introduction.
- Improve quality of figure 1
- Which standards were used for the casting process?
- Add images of tensile test samples before and after.
- Add images of tensile test setup and equipment.
- Section 3 rename to Results and Discussion
- The title of the manuscript is very long, please consider shortening it.
- Figure 3 add scale bar for each of the images.
- Again, please combine all small paragraphs into larger ones, there are so many everywhere in the manuscript. Anything less than 4-5 lines should be combined with previous or following paragraph.
- Figures 4,5 and 6 add some arrows and text to explain to the readers some of the interesting observations found in those images.
- Enlarge Figure 9
- What is the unit for the frequency in Figure 9?
- How many times was each tensile test repeated?
- Table 3 is better represented in bar chart for ease of readability.
- I think the novelty in this work is not clear, also the work done here is kind of very basic experimental level which does not show any attempt to compare the results with previous works related to it.
- The results are merely described and is limited to comparing the experimental observation and describing results. The authors are encouraged to include a more detailed results and discussion section and critically discuss the observations from this investigation with existing literature.
- Conclusion can be expanded or perhaps consider using bullet points (1-2 bullet points) from each of the subsections.
Author Response
- Please consider reviewing the abstract and highlight the novelty, major findings, and conclusions. I suggest reorganizing the abstract, highlighting the novelties introduced. The abstract should contain answers to the following questions:
- What problem was studied and why is it important?
- What methods were used?
- What conclusions can be drawn from the results? (Please provide specific results and not generic ones).
- The abstract must be improved. Please use numbers or % terms to clearly shows us the results in your experimental work. Please expand the abstract.
Answer: The abstract has been revised with comments above.
- Please consider reporting on studies related to your work from mdpi journals.
Answer: Authors have not yet published their works in mdpi journals.
- The introduction must be expanded, please consider improving the introduction, provide more in-depth critical review about past studies similar to your work, mention what they did and what were their main findings then highlight how does your current study brings new difference to the field.
Answer: The introduction has been revised with the Reviewer sugestions.
- The abstract should be on paragraph.
Answer: The abstract has been edited.
- Please combine small paragraphs into larger ones, I can see many of them in the manuscript especially in the introduction.
Answer: The text has been edited.
- Improve quality of figure 1 and 12. Add images of tensile test samples before and after.
Answer: The figure 1 has been changed.
- Which standards were used for the casting process?
Answer: Experimental castings were made of ductile cast iron according to standard PN-EN 1563:2018-10. The text was complete with this information.
- Add images of tensile test samples before and after.
Answer: Authors do not have samples anymore and it is not possibile to attach photo. Hovewer on figure 1 drawing and dimensions of circular cross-section test pieces for mechanical properties testing were attached.
- Add images of tensile test setup and equipment.
Answer: The strenght test methodology has been completed.
- Section 3 rename to Results and Discussion.
Answer: A change has been made.
- The title of the manuscript is very long, please consider shortening it.
Answer: A change has been made.
- Figure 3 add scale bar for each of the images.
Answer: Instead of the scale bar, there is information about the dimensions of the castings in the description of figure 3.
- Again, please combine all small paragraphs into larger ones, there are so many everywhere in the manuscript. Anything less than 4-5 lines should be combined with previous or following paragraph.
Answer: A change has been made.
- Figures 4,5 and 6 add some arrows and text to explain to the readers some of the interesting observations found in those images.
Answer: A change has been made.
- Enlarge Figure 9
Answer: A change has been made.
- What is the unit for the frequency in Figure 9?
Answer: The description of figure 9 has been completed.
- How many times was each tensile test repeated?
Answer: The text was complete with this information.
- Table 3 is better represented in bar chart for ease of readability.
Answer: A change has been made.
- I think the novelty in this work is not clear, also the work done here is kind of very basic experimental level which does not show any attempt to compare the results with previous works related to it.
Answer: The text was complete with this information.
- The results are merely described and is limited to comparing the experimental observation and describing results. The authors are encouraged to include a more detailed results and discussion section and critically discuss the observations from this investigation with existing literature.
Answer: The text was complete with this information.
- Conclusion can be expanded or perhaps consider using bullet points (1-2 bullet points) from each of the subsections.
Answer: A change has been made.

Reviewer 2 Report
This manuscript reports a study on the effect of casting molding materials on the microstructure and properties of casting iron. There are some interesting results, while a major revision on the presentation and discussion are suggested. Some comments and questions are listed below for consideration. 1. The title is too long. The novelty of the study should be presented more clearly. 2. What are the sample dimensions and strain rates for tensile testing? How many specimens did you tested for each casting? 3. In Table 3, what about the error bars of the strength and elongation? What is the meaning of "A"? 4. In Figs. 4-7, what are the two zones you mentioned, i.e., surface layer and the area of 10 mm from the surface? 5. What are the effects of these two zones on the mechanical properties? 6. In Fig. 8, comparing the graphite nodules morphology between casting No.2 and No. 3, i.e., (b) and (c), it seems there are no big difference. Why is it so different for the statistical results in Figs. 9(b) and (c)? 7. Can you explain more specifically for the origins of the difference in the microstructures of the matrix among different castings? 8. More discussions for the mechanical property results in Table 3 should be given. 9. In Conclusion Section, point #5 is more like a perspective rather than a conclusionAuthor Response
- The title is too long. The novelty of the study should be presented more clearly.
Answer: A change has been made.
- What are the sample dimensions and strain rates for tensile testing? How many specimens did you tested for each casting?
Answer: The text was complete with this information.
- In Table 3, what about the error bars of the strength and elongation? What is the meaning of "A"?
Answer: The text was complete with this information.
- In Figs. 4-7, what are the two zones you mentioned, i.e., surface layer and the area of 10 mm from the surface?
Answer: The text was complete with this information.
- What are the effects of these two zones on the mechanical properties?
Answer: The text was complete with this information.
- In Fig. 8, comparing the graphite nodules morphology between casting No.2 and No. 3, i.e., (b) and (c), it seems there are no big difference. Why is it so different for the statistical results in Figs. 9(b) and (c)?
Answer: The text was complete with this information.
- Can you explain more specifically for the origins of the difference in the microstructures of the matrix among different castings?
Answer: The text was complete with this information.
- More discussions for the mechanical property results in Table 3 should be given.
Answer: The text was complete with this information.
- In Conclusion Section, point #5 is more like a perspective rather than a conclusion.
Answer: A change has been made.

Reviewer 3 Report
This manuscript presents the results of investigations on a relevant subject matter of Journal «Materials». The manuscript deals with ductile cast iron modified with various organic compounds. The results are clearly stated in the manuscript, but their analysis is not enough. Authors should check the English text carefully.
Comments
(I) The title of manuscript should be changed. Since the role of polycaprolactone (Sample No.2) is remained unclear.
(II) The Introduction should be supplemented by references to until recently research by other scientists.
(III) Page 4 lines 137-138: Is this the composition of cast iron impurities? How much iron is there?
(IV) The authors concluded that organic additives affect the phase composition of cast iron castings. However, there is no XRD data confirming their phase composition. The microscope makes it possible to draw a conclusion only about the microstructure of the sample, and not about the phase composition.
(V) Figure 5d: Are the values on the y-axis increased correctly?
(VI) Table 3: How was the "A,%" parameter calculated? Formulas should be given in the Section 2. "Materials and Methods".
(VII) Section 3.3: The results should be analyzed in detail with a description of certain examples of the influence of the microstructure of the samples on the tensile of cast iron.
(VIII) Page 10 Lines 311-314: This thesis does not apply to the conclusion, it should be deleted.
I hope that my comments will be useful to the authors.
In the presented form, the article cannot be published in the Journal «Materials». I recommend this paper to be accepted for the publication with major revision.
Best regards, Reviewer
Author Response
- The title of manuscript should be changed. Since the role of polycaprolactone (Sample No.2) is remained unclear.
Answer: A change has been made. The text was complete with this information.
- The Introduction should be supplemented by references to until recently research by other scientists.
Answer: The text was complete with this information.
- Page 4 lines 137-138: Is this the composition of cast iron impurities? How much iron is there?
Answer: The text was complete with this information.
- The authors concluded that organic additives affect the phase composition of cast iron castings. However, there is no XRD data confirming their phase composition. The microscope makes it possible to draw a conclusion only about the microstructure of the sample, and not about the phase composition.
Answer: Authors meant the metal matrix phase composition as perlite or ferrite.
- Figure 5d: Are the values on the y-axis increased correctly?
Answer: The description of figure 9 has been completed.
- Table 3: How was the "A,%" parameter calculated? Formulas should be given in the Section 2. "Materials and Methods".
Answer: The text was complete with this information.
- Section 3.3: The results should be analyzed in detail with a description of certain examples of the influence of the microstructure of the samples on the tensile of cast iron.
Answer: The text was complete with this information.
- Page 10 Lines 311-314: This thesis does not apply to the conclusion, it should be deleted.
Answer: A change has been made.

Round 2
Reviewer 1 Report
Authors did not answer all comments raised in past review carefully, please check and update the manuscript
First of all, the authors must improve the introduction. please make it one large section and not divided into several ones.
Line 211 mention the full number of the standard instead of referencing it only.
Figure 3 add scale bar to the images at the bottom
Figure 9 add unit for frequency
Lines 357-370 combine into one larger paragraph
Author Response
Małgorzata Hosadyna-Kondracka, PhD
Łukasiewicz Research Network - Krakow Institute of Technology
Zakopiańska 73, 30-418 Krakow, Poland
+48 12 26 18 229
Editor-in-Chief
Materials
February 09, 2022
Dear Editor,
Thank you very much for the second review of the manuscript ID: materials-1551357 with a proposed new title “Effect of the biodegradable component addition to the molding sand on the microstructure and properties of ductile iron castings” by My, Katarzyna Major-Gabryś, Adelajda Polkowska and Małgorzata Warmuzek Name. I would like to thank once again Reviewers for valuable comments.
According to Reviewers recommendations the manuscript has been corrected. All the changes have been highlighted in the corrected version of the paper.
Responses to reviews:
Review 1 (Round 2)
1. Authors did not answer all comments raised in past review carefully, please check and update the manuscript
Answer: The made changes are marked below in blue.
Review 1 (Round 1)
- Please consider reviewing the abstract and highlight the novelty, major findings, and conclusions. I suggest reorganizing the abstract, highlighting the novelties introduced. The abstract should contain answers to the following questions:
- What problem was studied and why is it important? ,,A novelty is the use of molding sand with a two-component binder: furfuryl resin - polycaprolactone PCL biomaterial.” (line 19-21)
- What methods were used? ,,The molds were poured with ductile iron according to standard PN-EN 1563:2018-10. The microstructure of the experimental castings was examined on metallographic cross-sections with PN-EN ISO 945-1:2019-09 standard. Observations were made in the area at the casting/mold boundary and in a zone approximately 10 mm from the surface of the casting with light microscope. The tensile test at room temperature was conducted according to standard PN-EN ISO 6892-1:2016-09. Circular cross-section test pieces, machined from samples taken from castings, were used” (line 31-36)
- What conclusions can be drawn from the results? (Please provide specific results and not generic ones). ,,the use of molding sand with furfuryl resin with the addition of biodegradable PCL material does not lead to an unfavorable modification of the microstructure and mechanical properties in the casting” (line 43-45)
- The abstract must be improved. Please use numbers or % terms to clearly shows us the results in your experimental work. Please expand the abstract. ,,The samples taken from the Casting no. 2, made on the acidic molding sand with the participation of biodegradable material, had the highest average strength UTS - among all tested molding sands - of 672 MPa. However, the elongation after fracture was 48% lower compared to the reference samples from the Casting no. 1 from the sand without the addition of PCL.” (line 45-49)
Answer: The abstract has been revised with comments above.
- Please consider reporting on studies related to your work from mdpi journals.
Answer: Authors have not published yet their works in mdpi journals.
- The introduction must be expanded, please consider improving the introduction, provide more in-depth critical review about past studies similar to your work, mention what they did and what were their main findings then highlight how does your current study brings new difference to the field.
Answer: The introduction has been revised with the Reviewer suggestions.
- The abstract should be on paragraph.
Answer: The abstract has been edited.
- Please combine small paragraphs into larger ones, I can see many of them in the manuscript especially in the introduction.
Answer: The text has been edited.
- Improve quality of figure 1.
Answer: The figure 1 has been changed.
- Which standards were used for the casting process?
Answer: Experimental castings were made of ductile cast iron according to standard PN-EN 1563:2018-10. The text was completed with this information.
- Add images of tensile test samples before and after.
Answer: Authors do not have samples anymore and it is not possible to attach photo. However on figure 1 drawing and dimensions of circular cross-section test pieces for mechanical properties testing were attached.
- Add images of tensile test setup and equipment.
Answer: The strength test methodology has been completed. ,,The tensile test at room temperature was conducted according to standard PN-EN ISO 6892-1:2016-09, method B [24]. Circular cross-section test pieces of 14 mm diameter (drawing and dimensions on Figure 1), machined from samples taken from castings, were used. 3 pieces were tested for each casting. Examinations were carried out on EU-20 strength testing machine with a range of 0-200 kN. The strain rate for tensile testing was 16 MPa/s. The percentage elongation after fracture A was calculated from the Formula 1 [24]:” (line 237-246)
- Section 3 rename to Results and Discussion.
Answer: The change has been made.
- The title of the manuscript is very long, please consider shortening it.
Answer: The title has been changed.
- Figure 3 add scale bar for each of the images.
Answer: Instead of the scale bar, there is information about the dimensions of the castings in the description of figure 3. The scale bar was added to Figure 3.
- Again, please combine all small paragraphs into larger ones, there are so many everywhere in the manuscript. Anything less than 4-5 lines should be combined with previous or following paragraph.
Answer:The change has been made.
- Figures 4,5 and 6 add some arrows and text to explain to the readers some of the interesting observations found in those images.
Answer: The change has been made.
- Enlarge Figure 9
Answer: The change has been made.
- What is the unit for the frequency in Figure 9?
Answer: The description of axis on figure 9 has been completed. Frequency is the number of occurrences of a repeating event per unit of time or in a range (class) of other variable ( e.g. histogram or normal distribution).
- How many times was each tensile test repeated?
Answer: The text was completed with this information.
- Table 3 is better represented in bar chart for ease of readability.
Answer: The change has been made.
- I think the novelty in this work is not clear, also the work done here is kind of very basic experimental level which does not show any attempt to compare the results with previous works related to it.
Answer: The text was completed with this information in abstract and introduction.
- The results are merely described and is limited to comparing the experimental observation and describing results. The authors are encouraged to include a more detailed results and discussion section and critically discuss the observations from this investigation with existing literature.
Answer: The text was completed with this information.
- Conclusion can be expanded or perhaps consider using bullet points (1-2 bullet points) from each of the subsections.
Answer: The change has been made.
Review 1 (Round 2)
2. First of all, the authors must improve the introduction. Please make it one large section and not divided into several ones.
Answer: The introduction was edited to one large section.
3. Line 211 mention the full number of the standard instead of referencing it only.
Answer: The change has been made.
4. Figure 3 add scale bar to the images at the bottom
Answer: The scale bar was added to Figure 3.
5. Figure 9 add unit for frequency
Answer: The description of axis on figure 9 has been completed.
6. Lines 357-370 combine into one larger paragraph
Answer: The change has been made.
Review 2 (Round 2)
The authors have revised some places, while the response letter was not clearly written. Thus, I cannot see the answers for the questions #5-#7 proposed in the first round review. Please consider to revise the manuscript and the response letter again.
- The title is too long. The novelty of the study should be presented more clearly.
Answer: The change has been made.
- What are the sample dimensions and strain rates for tensile testing? How many specimens did you tested for each casting?
Answer: The text was completed with this information.
- In Table 3, what about the error bars of the strength and elongation? What is the meaning of "A"?
Answer: The text was completed with this information.
- In Figs. 4-7, what are the two zones you mentioned, i.e., surface layer and the area of 10 mm from the surface?
Answer: The text was completed with this information.
- What are the effects of these two zones on the mechanical properties?
Answer: The text was completed with this information. The effects of superficial zone on mechanical properties can be considered in two aspects, related to: a. Material and b. Cast part.
- a. Material effects result from the diffusive exchange of elements at the metal/form interface. The resulting concentration gradient of individual components in the liquid alloy affects the final image of the microstructure of the material. This impact concerns both the matrix composition and graphite morphology, and then the observed differentiation of properties (Tables 2 and 3).
- Different range of superficial zone of different matrix and graphite morphology observed in the examined cast parts can influence on useful properties of cast part during specific exploitation conditions. (line 420-431).
- In Fig. 8, comparing the graphite nodules morphology between casting No.2 and No. 3, i.e., (b) and (c), it seems there are no big difference. Why is it so different for the statistical results in Figs. 9(b) and (c)?
Answer: The text was completed with this information. Quantitative image analysis revealed some differences in graphite morphology, not always accounted for in the visual analysis results. From a comparison of the histograms in Fig. 9b and b, it follows that the spheroidal size distribution is bimodal, i.e. there are probably two sets of nodules: in sample 2 (Fig. 9b) the first with a diameter of 120-140 mm and the second with a diameter of 30-50 mm, while in sample 3 (Fig. 9c) - the first with a diameter of 90-100 mm and the second, probably not very numerous, with a diameter of about 130 mm. This difference in the size distribution of spheroids can be considered a factor of the synergy effect of both components of the microstructure, i.e. graphite and matrix, on the mechanical properties (Table 3). (line 453-462)
- Can you explain more specifically for the origins of the difference in the microstructures of the matrix among different castings?
Answer: The text was completed with this information. At this stage of the research, it can be assumed that the difference in the matrix microstructure results indirectly from:
- Direct diffusion interactions at the metal /mold interface, which are influenced by the composition of the molding sand,
- The temperature field occurring in the casting at the subsequent stages of solidification, when the microstructure is formed, determined by the thermophysical properties of the mold depending on the components used. As this presented results concern of initial stage of research, these phenomena are still under examination. (line 463-471)
- More discussions for the mechanical property results in Table 3 should be given.
Answer: The text was completed with this information.
- In Conclusion Section, point #5 is more like a perspective rather than a conclusion.
Answer: The change has been made.
Review 3 (Round 2)
I consider that the paper has been improved according to Reviewer’s recommendations. I recommend this paper to be accepted for the publication in Journal «Materials».
«But»!!!
(I) Equation 1: Percentage as a unit of measure must be specified.
Answer: The Equation 1 was completed with a percentage unit.
(II) Page 14 lines 435-438: This thesis does not apply to the conclusion, it should be deleted.
Answer: This thesis was deleted.
Amendments made after the second round of reviews are highlighted in green in the manuscript text.
I believe that the revised manuscript is appropriate for publication by the Materials journal as a research paper.
Thank you for your consideration!
Sincerely,
Małgorzata Hosadyna-Kondracka

Reviewer 2 Report
The authors have revised some places, while the response letter was not clearly written. Thus, I cannot see the answers for the questions #5-#7 proposed in the first round review. Please consider to revise the manuscript and the response letter again.
Author Response

(The authors gave the same response as above.)

Reviewer 3 Report
I consider that the paper has been improved according to Reviewer’s recommendations. I recommend this paper to be accepted for the publication in Journal «Materials».
«But»!!!
(I) Equation 1: Percentage as a unit of measure must be specified.
(II) Page 14 lines 435-438: This thesis does not apply to the conclusion, it should be deleted.
Best regards, Reviewer
Author Response
Małgorzata Hosadyna-Kondracka, PhD
Łukasiewicz Research Network - Krakow Institute of Technology
Zakopiańska 73, 30-418 Krakow, Poland
+48 12 26 18 229
Editor-in-Chief
Materials
February 09, 2022
Dear Editor,
Thank you very much for the second review of the manuscript ID: materials-1551357 with a proposed new title “Effect of the biodegradable component addition to the molding sand on the microstructure and properties of ductile iron castings” by My, Katarzyna Major-Gabryś, Adelajda Polkowska and Małgorzata Warmuzek Name. I would like to thank once again Reviewers for valuable comments.
According to Reviewers recommendations the manuscript has been corrected. All the changes have been highlighted in the corrected version of the paper.
Responses to reviews:
Review 1 (Round 2)
1. Authors did not answer all comments raised in past review carefully, please check and update the manuscript
Answer: The made changes are marked below in blue.
Review 1 (Round 1)
- Please consider reviewing the abstract and highlight the novelty, major findings, and conclusions. I suggest reorganizing the abstract, highlighting the novelties introduced. The abstract should contain answers to the following questions:
- What problem was studied and why is it important? ,,A novelty is the use of molding sand with a two-component binder: furfuryl resin - polycaprolactone PCL biomaterial.” (line 19-21)
- What methods were used? ,,The molds were poured with ductile iron according to standard PN-EN 1563:2018-10. The microstructure of the experimental castings was examined on metallographic cross-sections with PN-EN ISO 945-1:2019-09 standard. Observations were made in the area at the casting/mold boundary and in a zone approximately 10 mm from the surface of the casting with light microscope. The tensile test at room temperature was conducted according to standard PN-EN ISO 6892-1:2016-09. Circular cross-section test pieces, machined from samples taken from castings, were used” (line 31-36)
- What conclusions can be drawn from the results? (Please provide specific results and not generic ones). ,,the use of molding sand with furfuryl resin with the addition of biodegradable PCL material does not lead to an unfavorable modification of the microstructure and mechanical properties in the casting” (line 43-45)
- The abstract must be improved. Please use numbers or % terms to clearly shows us the results in your experimental work. Please expand the abstract. ,,The samples taken from the Casting no. 2, made on the acidic molding sand with the participation of biodegradable material, had the highest average strength UTS - among all tested molding sands - of 672 MPa. However, the elongation after fracture was 48% lower compared to the reference samples from the Casting no. 1 from the sand without the addition of PCL.” (line 45-49)
Answer: The abstract has been revised with comments above.
- Please consider reporting on studies related to your work from mdpi journals.
Answer: Authors have not published yet their works in mdpi journals.
- The introduction must be expanded, please consider improving the introduction, provide more in-depth critical review about past studies similar to your work, mention what they did and what were their main findings then highlight how does your current study brings new difference to the field.
Answer: The introduction has been revised with the Reviewer suggestions.
- The abstract should be on paragraph.
Answer: The abstract has been edited.
- Please combine small paragraphs into larger ones, I can see many of them in the manuscript especially in the introduction.
Answer: The text has been edited.
- Improve quality of figure 1.
Answer: The figure 1 has been changed.
- Which standards were used for the casting process?
Answer: Experimental castings were made of ductile cast iron according to standard PN-EN 1563:2018-10. The text was completed with this information.
- Add images of tensile test samples before and after.
Answer: Authors do not have samples anymore and it is not possible to attach photo. However on figure 1 drawing and dimensions of circular cross-section test pieces for mechanical properties testing were attached.
- Add images of tensile test setup and equipment.
Answer: The strength test methodology has been completed. ,,The tensile test at room temperature was conducted according to standard PN-EN ISO 6892-1:2016-09, method B [24]. Circular cross-section test pieces of 14 mm diameter (drawing and dimensions on Figure 1), machined from samples taken from castings, were used. 3 pieces were tested for each casting. Examinations were carried out on EU-20 strength testing machine with a range of 0-200 kN. The strain rate for tensile testing was 16 MPa/s. The percentage elongation after fracture A was calculated from the Formula 1 [24]:” (line 237-246)
- Section 3 rename to Results and Discussion.
Answer: The change has been made.
- The title of the manuscript is very long, please consider shortening it.
Answer: The title has been changed.
- Figure 3 add scale bar for each of the images.
Answer: Instead of the scale bar, there is information about the dimensions of the castings in the description of figure 3. The scale bar was added to Figure 3.
- Again, please combine all small paragraphs into larger ones, there are so many everywhere in the manuscript. Anything less than 4-5 lines should be combined with previous or following paragraph.
Answer:The change has been made.
- Figures 4,5 and 6 add some arrows and text to explain to the readers some of the interesting observations found in those images.
Answer: The change has been made.
- Enlarge Figure 9
Answer: The change has been made.
- What is the unit for the frequency in Figure 9?
Answer: The description of axis on figure 9 has been completed. Frequency is the number of occurrences of a repeating event per unit of time or in a range (class) of other variable ( e.g. histogram or normal distribution).
- How many times was each tensile test repeated?
Answer: The text was completed with this information.
- Table 3 is better represented in bar chart for ease of readability.
Answer: The change has been made.
- I think the novelty in this work is not clear, also the work done here is kind of very basic experimental level which does not show any attempt to compare the results with previous works related to it.
Answer: The text was completed with this information in abstract and introduction.
- The results are merely described and is limited to comparing the experimental observation and describing results. The authors are encouraged to include a more detailed results and discussion section and critically discuss the observations from this investigation with existing literature.
Answer: The text was completed with this information.
- Conclusion can be expanded or perhaps consider using bullet points (1-2 bullet points) from each of the subsections.
Answer: The change has been made.
Review 1 (Round 2)
2. First of all, the authors must improve the introduction. Please make it one large section and not divided into several ones.
Answer: The introduction was edited to one large section.
3. Line 211 mention the full number of the standard instead of referencing it only.
Answer: The change has been made.
4. Figure 3 add scale bar to the images at the bottom
Answer: The scale bar was added to Figure 3.
5. Figure 9 add unit for frequency
Answer: The description of axis on figure 9 has been completed.
6. Lines 357-370 combine into one larger paragraph
Answer: The change has been made.
Review 2 (Round 2)
The authors have revised some places, while the response letter was not clearly written. Thus, I cannot see the answers for the questions #5-#7 proposed in the first round review. Please consider to revise the manuscript and the response letter again.
- The title is too long. The novelty of the study should be presented more clearly.
Answer: The change has been made.
- What are the sample dimensions and strain rates for tensile testing? How many specimens did you tested for each casting?
Answer: The text was completed with this information.
- In Table 3, what about the error bars of the strength and elongation? What is the meaning of "A"?
Answer: The text was completed with this information.
- In Figs. 4-7, what are the two zones you mentioned, i.e., surface layer and the area of 10 mm from the surface?
Answer: The text was completed with this information.
- What are the effects of these two zones on the mechanical properties?
Answer: The text was completed with this information. The effects of superficial zone on mechanical properties can be considered in two aspects, related to: a. Material and b. Cast part.
a. Material effects result from the diffusive exchange of elements at the metal/form interface. The resulting concentration gradient of individual components in the liquid alloy affects the final image of the microstructure of the material. This impact concerns both the matrix composition and graphite morphology, and then the observed differentiation of properties (Tables 2 and 3).
b. Different range of superficial zone of different matrix and graphite morphology observed in the examined cast parts can influence on useful properties of cast part during specific exploitation conditions. (line 420-431).
6. In Fig. 8, comparing the graphite nodules morphology between casting No.2 and No. 3, i.e., (b) and (c), it seems there are no big difference. Why is it so different for the statistical results in Figs. 9(b) and (c)?
Answer: The text was completed with this information. Quantitative image analysis revealed some differences in graphite morphology, not always accounted for in the visual analysis results. From a comparison of the histograms in Fig. 9b and b, it follows that the spheroidal size distribution is bimodal, i.e. there are probably two sets of nodules: in sample 2 (Fig. 9b) the first with a diameter of 120-140 mm and the second with a diameter of 30-50 mm, while in sample 3 (Fig. 9c) - the first with a diameter of 90-100 mm and the second, probably not very numerous, with a diameter of about 130 mm. This difference in the size distribution of spheroids can be considered a factor of the synergy effect of both components of the microstructure, i.e. graphite and matrix, on the mechanical properties (Table 3). (line 453-462)
- Can you explain more specifically for the origins of the difference in the microstructures of the matrix among different castings?
Answer: The text was completed with this information. At this stage of the research, it can be assumed that the difference in the matrix microstructure results indirectly from:
a. Direct diffusion interactions at the metal /mold interface, which are influenced by the composition of the molding sand,
b. The temperature field occurring in the casting at the subsequent stages of solidification, when the microstructure is formed, determined by the thermophysical properties of the mold depending on the components used. As this presented results concern of initial stage of research, these phenomena are still under examination. (line 463-471)
8. More discussions for the mechanical property results in Table 3 should be given.
Answer: The text was completed with this information.
- In Conclusion Section, point #5 is more like a perspective rather than a conclusion.
Answer: The change has been made.
Review 3 (Round 2)
I consider that the paper has been improved according to Reviewer’s recommendations. I recommend this paper to be accepted for the publication in Journal «Materials».
«But»!!!
(I) Equation 1: Percentage as a unit of measure must be specified.
Answer: The Equation 1 was completed with a percentage unit.
(II) Page 14 lines 435-438: This thesis does not apply to the conclusion, it should be deleted.
Answer: This thesis was deleted.
Amendments made after the second round of reviews are highlighted in green in the manuscript text.
I believe that the revised manuscript is appropriate for publication by the Materials journal as a research paper.
Thank you for your consideration!
Sincerely,
Małgorzata Hosadyna-Kondracka

Round 3
Reviewer 1 Report
The authors should consider improving the quality of their graphs and figures. they have poor resolution
The authors provided the answers to the comments from the second round of review and made sufficient changes in the manuscript according to these comments. I recommend this manuscript for a publication in its present form.
Author Response
According to the Reviewer suggestion the quality of Figures: 1, 2, 3, 9 and Table 3 has been improved by increasing their resolution. These changes have been highlighted in the corrected version of the paper.
I would like to thank Reviewer for all comments and for recommending the manuscript for publication.
Reviewer 2 Report
I have no additional comments on this manuscript.
Author Response
I would like to thank Reviewer for all comments.